# Stepwise Approach for Tracing the Geographical Origins of the Manila Clam *Ruditapes philippinarum* Using Dual-Element Isotopes and Carbon Isotopes of Fatty Acids

**DOI:** 10.3390/foods11131965

**Published:** 2022-07-01

**Authors:** Young-Shin Go, Eun-Ji Won, Seung-Hee Kim, Dong-Hun Lee, Jung-Ha Kang, Kyung-Hoon Shin

**Affiliations:** 1Department of Marine Sciences and Convergent Technology, Hanyang University, Ansan 15588, Korea; hc12sook@gmail.com (Y.-S.G.); bearpig4528@gmail.com (S.-H.K.); ldh301@korea.kr (D.-H.L.); 2Marine Environmental Management Division, National Institute of Fisheries Science, Busan 46083, Korea; 3Institute of Marine & Atmospheric Sciences, Hanyang University, Ansan 15588, Korea; ejwon@hanyang.ac.kr; 4Aquaculture Industry Division, West Sea Fisheries Research Institute, National Institute of Fisheries Science, Incheon 22383, Korea; genetics@korea.kr

**Keywords:** authentication, manila clam, compound-specific isotope analysis, fatty acids, dual-isotopes, linear discriminant analysis

## Abstract

While there are many studies that have reported methods for tracing the geographical origin of seafoods, most of them have focused on identifying parameters that can be used effectively and not the direct application of these methods. In this study, we attempted to differentiate the geographical origins of the Manila clam *R. philippinarum* collected from different sites in Korea, the Democratic People’s Republic of Korea, and China using a combination of analyses based on dual-element isotopes, fatty acids (FAs), and compound-specific isotopic analysis of FAs. We hypothesized that a stepwise application of new parameters to unclassified samples could achieve this objective by integrating new information while reducing time and labor. The FA profiles and compound-specific carbon isotopic values of FAs were found to enhance the discrimination power of determining the geographic origin up to 100%. Our findings demonstrate the advantageousness of using several parameters simultaneously over the conventional method of employing individual analytical methods when identifying geographic origins of the Manila clam, which could have implications for tracing the origins of different shellfish species or other food products as well.

## 1. Introduction

The geographical origin discrimination of foods is a prominent issue in the food industry [1,2] that has recently been considered to be an important factor as it demonstrates not only the origin of a food but also the credit [3,4]. In particular, efforts for determining the geographical origins of seafood are lagging. Much of the difficulty in tracing the origin of seafood items derives from supply chain complexity [4,5]. The supply chain is also often hindered by several wild-caught seafood products, which, amidst the increase in consumer demands, resource limitations, and high value, only contributes to the sourcing difficulties.

Recently, seafood, with the potential for large economic profit, has also become a factor of certain global issues, such as varying prices of products from special localities (e.g., specialties) [4,6]. This arises especially between farmed/wild or imported/native products [6,7]. In Korea, the safety of fishery products has been garnering more concern, along with issues such as the intentional and unintended contamination of seafood imported from other countries [8]. With environmental pollution from heavy metals or radioactive materials affecting people’s health, there is a need for geographical origin information to address or prevent these health problems, rather than for basic information, such as the location where the seafood was fished, farmed, and packed. [9]. Furthermore, although the government has set regulations for tracking products through the supply chain, illegal acts (e.g., fraud), including those related to the origins of seafood, occur frequently, which affect the fishery markets and have negative economic and social impacts [4]. The Manila clam *Ruditapes philippinarum*, for example, is a valuable commercial bivalve with a production rate of approximately 3.9 million tons per year [10]—just below that of oysters and mussels. Fraudulent acts surrounding these clams often occur because the preference for domestic clams has led to their high prices in Korea [11]. The geographical origins of clams were initially distinguished based on morphological differences, which is a simple approach, and DNA sequencing was suggested as a promising approach for differentiation based on phylogenetic similarities [12]. However, genetic markers can no longer be a good basis for differentiation of geographical origin in Korea, as more than 4000 tons of Manila clam seeds have been imported from China and cultivated in coastal areas [8]. Korea Institute of Marine Science and Technology Promotion [8] demonstrated that, since 2009, most Manila clams in Korea have the same DNA sequences as those of Manila clams in China. Thus, geographical authentication requires markers or endpoints for geographical information that indicate the regions where the organisms were grown or cultured. Similarly, for beef and certain farmed fish species (*Misgurnus mizolepis*), the geographical origin (country) on the labels is determined based on whether they have lived (or were cultivated) in the imported country for a specified period after importation according to the tracing system [13].

The increasing demand for traceability has led to various studies that have presented scientific evidence for some techniques as well as for national policies [4,7]. For agricultural products, dairy products, and processed foods, such as oil and wine, reliable results have been reported for several approaches using stable isotopes, DNA, metabolomics, and microbiological information [14,15]. For example, stable isotope analysis (SIA) is an effective method for determining environmental information, habitat, and ecosystem-based diets [16,17]. As a powerful method for providing information on sources, processes, and fluxes with ratios (e.g., ^13^C/^12^C, ^15^N/^14^N), SIA has been applied in diverse research fields, including those of ecology and environmental sciences. The use of stable isotopic values (C and N) in organisms can reveal the energy sources (organic sources) and their movement through the food web based on kinetic reactions (e.g., fractionation factor) [18,19]. It can also be adopted for geographical source tracing, as differences in sources and changes caused by specific processes can provide useful information for distinguishing groups of products based on geographical differences [2,19,20]. Thus, this approach has been used for tracing the raw-material sources of products, such as juice, wine, milk, honey, oil, and meat [1]. Watanabe et al. [21] showed the possibility of adopting this approach for seafood as the composition of stable nitrogen isotopes in shellfish varies according to the environment, even within the same species. Dang et al. [22] showed that the patterns of stable isotopes (e.g., carbon and nitrogen) measured in clams and environmental samples, such as particulate organic matter (POM) were consistent with the spatial gradients derived from their geological and artificial (land-use) characteristics at the study sites. Furthermore, several studies have followed and applied multi-element isotopes (C, N, S) for determining geographical origin because the combination of elements can provide comprehensive information on biological and environmental differences [23,24].

Another approach for determining the origin of organic matter (e.g., diet) in marine environments is based on fatty acid (FA) composition [2,25]. Because FAs in the body vary depending on diet, they can be used as reliable indicators of diet in specific species or groups in an ecosystem. For example, farmed fish showed a considerably higher content of lipids and n-6 FA (18:2n-6) levels, whereas wild fish exhibited relatively low FA content, high levels of n-3 FAs, and a high ratio of n-3 to n-6 FAs [26], therefore indicating dietary differences. Additionally, certain FAs originate from primary producers because the body of consumers cannot synthesize them [27,28,29]. For example, Volkman et al. [30] showed a clear distinction of FA patterns among four different producers (microalgae, bacteria, fungi, and higher plants), which were reflected directly in the higher-trophic-level organisms. Manila clams can serve as a candidate for the determination of geographical origin based on FA profile as they are filter feeders and are influenced by biological sources (e.g., plankton) [22]; thus, we expect that the FAs in clams indicate differences in the plankton community, which may vary regionally [31]. In addition, compound-specific isotope analysis (CSIA), which has recently been gaining attention, can provide additional specific information on the habitat- and diet-derived FA patterns and carbon sources of an organism (e.g., δ^13^C-FA) [32].

Considering the demand and interest regarding the tracing of origin using habitat information, we attempted to differentiate the geographical origins of Manila clam *R. philippinarum* collected from different sites using a combination of analyses based on multi-element isotopes, FAs, and CSIA of FAs. We hypothesized that a stepwise application of new parameters to unclassified samples could help achieve this aim by integrating new information while reducing the consumption of time and labor. First, samples from different sites in Korea (Section 2.1) were collected; second, we adopted the stepwise approach for selected samples from two adjacent countries (China and the Democratic People’s Republic of Korea (DPR Korea)). As a preliminary study, each step was applied sequentially, taking into consideration the time required for the experiment and the technical accessibility, which was judged based on previous studies [19] (Figure 1). Finally, we determined the feasibility of this approach by evaluating whether it yields a discriminant rate greater than 97%, which is the rate achieved in similar studies that have used multi-isotope analysis for authenticating the geographical origin of seafoods [19,33].

## 2. Materials and Methods

### 2.1. Sample Collection

Clams from China and the DPR Korea with clearly proven origin information were used to verify the import procedure as per the National Institute of Fisheries Sciences. The Korean samples (*n* = 102) were collected from 17 sites in 2015 and 2016: Chungcheong (CC; *n* = 25), Jeonbuk (JB; *n* = 25), Jeonnam (JN; *n* = 26), Gyeongnam (GN; *n* = 26) in Korea, DPR Korea (NK; *n* = 16), and China (CHN; *n* = 15) (Figure 2). Samples were directly collected from each region, and the exact sampling site and date are listed in Table 1. After collecting samples from the sites, the samples were brought to the laboratory, and the adductor muscle was dissected and used as the target organ for isotope analysis to obtain long-term dietary information. The adductor muscle was selected because it has a slower turnover and much lower lipid content than other tissues [22]. The dissected samples were stored in a deep freezer (−80 °C) prior to homogenization using a mortar and pestle.

### 2.2. Stable Isotope Analysis of Bulk Samples

Approximately 1.0 mg of dried sample was used for nitrogen isotope (δ^15^N) analysis without further processing. For carbon isotope analysis, however, each homogenized sample was subjected to acidification to remove any inorganic carbon, which resulted in enriched isotopic values [34]. Additionally, lipids, which exhibit depleted isotopic values, were removed using a mixture of chloroform and methanol (2:1, *v/v* ratio)**,** as described in Kim et al. [20], where a mixed solvent was added to the sample and shaken to remove the lipid layer separated on the supernatant; this step was repeated thrice.

The δ^13^C and δ^15^N values of each sample were measured using an elemental analyzer (Vario PYRO cube, Elemental Analyzer GmbH., Langenselbold, Germany) combined with an isotope–ratio mass spectrometer (Isoprime 100, Isoprime Ltd., Cheadle, UK). The analytical precision was ± 0.3‰ for stable carbon isotopes by certified standards (IAEA-CH-3, δ^13^C = −24.72‰) and ± 0.3‰ for nitrogen isotopes by running standards (IAEA-N-1, δ^15^N = 0.4‰).

Stable isotope ratios were expressed using the delta (δ) notation. The delta (δ) notation represents the measurements relative to the standard during the actual analysis (Equation (1)).
(1)$$\delta \;\mathrm{Element}\;\left(‰\right)=\left(\frac{R_{\mathrm{sample}}}{R_{\mathrm{standard}}}-1\right)\times 1000$$
where R indicates the ratio of heavy (^13^C and ^15^N) and light elements (^12^C and ^14^N). Vienna Pee Dee Belemnite and atmospheric N_2_ were used as δ^13^C and δ^15^N references, respectively.

### 2.3. FA Content and CSIA of Phospholipid FA

Fifty-one samples (triplicates from 17 sites) were selected from the unclassified samples using linear discriminant analysis (LDA). Total FAs were extracted by an ultrasonic extraction method using solvent mixtures. Dichloromethane (DCM)/methanol (MeOH) (2/1, *v/v*) was added to the samples, and three ultrasonic extractions were repeated three times with the internal standard (i.e., nonadecanoic acid). The total lipid extracts were passed through a Na_2_SO_4_ column to remove moisture and dried under N_2_ gas. Subsequently, we divided the samples into neutral, apolar, and polar lipids (Appendix A) for further analysis of the content and carbon isotope composition of individual FAs. The experimental procedure is described in the Appendix A [35].

The polar fraction was reacted with 14% BF_3_ in MeOH at 60 °C for 10 min to yield FA methyl esters. After cooling, the methylated FAs were separated into phospholipid FA (PLFA) fractions on an SiO_2_ column using ethyl acetate solvent. The PLFAs were divided into three aliquots for analyzing the content and δ^13^C of the FAs. The first aliquot was analyzed for the concentration and composition of PLFAs. The concentration of PLFAs was determined by gas chromatography (GC) with flame-ionization detection using a DB-5 silica capillary column (60 m × 0.25 mm, 0.25 μm, Agilent Technologies, Santa Clara, CA, USA) in an Agilent 7890 Series in the splitless mode with helium as the carrier gas. The PLFA composition was measured by GC–mass spectrometry (GC–MS) using an Agilent GC interfaced with 5977B MS equipped with a HP-5MS column (25 m × 0.25 mm, 0.10 μm, Agilent Technologies). Details of the analytical conditions are described in the Appendix A. The second aliquot was analyzed for the double-bond positions in the PLFAs, which were determined by performing dimethyl disulfide analysis, according to a previously described procedure [36]. The third aliquot was further purified over Ag^2+^ impregnated silica columns using hexane/DCM (1/1, *v/v*), hexane/ethyl acetate (4/1, *v/v*), and acetone for separating saturated and unsaturated (monounsaturated and polyunsaturated) fractions. The stable carbon isotope values of PLFA (δ^13^C PLFAs) were determined using GC combustion isotope ratio mass spectrometry (GC-C-IRMS). An isotope ratio mass spectrometer (isoprime visION, Elementar, Germany) was coupled to a gas chromatograph (Hewlett Packard 7890 N series, Agilent Technologies) via a combustion interface (glass tube packed with copper oxide (CuO), operated at 850 °C). The samples were subjected to the same analysis conditions described for the GC–MS analyses; the detailed conditions are described in the Appendix A.

### 2.4. Statistical Analysis: Feature Scaling and LDA

Data are presented as the mean ± standard deviation (S.D). Before statistical analysis, the normal distribution of data and homogeneity of variances were evaluated using the Levene’s test. Significant differences in the values of each isotope or FA profiles according to region or country were analyzed using one-way analysis of variance (ANOVA) and post hoc analysis following Tukey’s test (*p* < 0.05). For additional statistical analysis, LDA, a feature-scaling method, was used for eliminating the errors generated in different values, signs, and variabilities from each element. Feature scaling is a method of standardizing the data used for statistical processing to values within the desired range. Finally, the data were normalized in the range of 0 to 1 using the following equation:(2)$${\mathrm{Ax}}^{\prime }=\frac{\mathrm{Ax}-\min\left(A\right)}{\max\left(A\right)-\min\left(A\right)}$$
where A: elements or FAs, x′: normalized values, x: stable isotope composition or FA composition; max(A): maximum values of stable isotope composition or FA composition; and min(A): minimum values of stable isotope composition or FA composition.

After normalizing the data using the feature scaling method, LDA was performed using the R studio program. LDA is a discriminant approach that attempts to model the differences among samples assigned to certain groups. Additionally, all LDA results were evaluated using a cross-validation step following the internal leave-one-out cross-validation (LOOCV) method [19].

## 3. Results and Discussions

### 3.1. First Step: A Dual-Isotope Analysis for Distinct Geographical Origins of Clams from Four Different Provinces (Coasts) of Korea

Manila clams sampled in Korea can be classified according to their geographical origins as clams from the northwestern (Chungcheong; CC), southwestern (Jeonbuk; JB), southwestern (Jeonnam; JN), and southeastern (Gyeongnam; GN) coastal areas. The stable isotopic values of the Manila clams collected from the four regions (Chungcheong, Jeonbuk, Jeonnam, and Gyeongnam) are listed in Table 1. The δ^13^C values of these four regions were −17.98 ± 0.44‰, −16.93 ± 0.41‰, −16.78 ± 0.31‰, and −16.13 ± 0.55‰, respectively. Generally, the geographical differences in the carbon isotope ratios indicate differences in dietary sources according to habitat, because the δ^13^C values indicate the organic matter sources via low fractionation of the consumer’s diet [16,17]. In this study, the δ^13^C values of clams may have reflected the POM, which could be related to the dietary source of the clam, as they are filter feeders [37,38]. In a study conducted in the Hichirippu Lagoon of Japan, Komorita et al. [39] demonstrated that the δ^13^C values of bivalves reflect dietary sources (such as POM). Our results of the samples from Korea were similar to those from previous studies. The δ^13^C values for Manila clams from Gwangyang and Jinju Bay, located in the Southern Sea of Korea, were −16.5 ± 0.4‰ and −16.9 ± 0.8‰, respectively [40,41]. Furthermore, the carbon isotope ratios measured in samples from JN (−17.1 to −16.5‰) were consistent with those in samples from Hwayang and Dolsan (−17.1 to –16.3‰), which were the sites of a previous study [19]. From a geographical point of view, the δ^13^C values of POM tended to increase from north to south in the Western Sea in Korea. In our previous study, however, Suh and Shin [37] demonstrated changes in the proportion of food sources from benthic organic matters to POM, resulting from seasonal variations. This result indicates that identification of a temporal range may be difficult by simply dividing regions according to δ^13^C values and that the integrated results of habitat and dietary effects would be effective in determining geographical differences in clams using a statistical analysis method, such as LDA.

In general, nitrogen isotopic compositions are more fractionated than carbon isotopic compositions during dietary assimilation, with an increase of a 2–4‰ increase in isotope enrichment at each trophic level, rendering it useful for indicating the trophic levels of organisms in their habitats [16]. The δ^15^N values of aquatic organisms can serve as a good index for understanding food webs because they depend on sources, dietary shifts, and vertical relations at the trophic level [16,17]. In particular, δ^15^N values reflect not only the variations at the baseline-level of food webs but also the differences in trophic relations in ecosystems [42,43,44]. For example, in coastal areas, the δ^15^N values of dissolved inorganic nitrogen (DIN) passing from rivers can affect the nitrogen isotopes in primary producers and subsequently in consumers [44]. Fukumori et al. [45] demonstrated that the δ^15^N of bivalves varied considerably among sites, reflecting the baseline values of δ^15^N in coastal areas. This indicates that δ^15^N values in clams could serve as a distinguishable parameter for tracing the geographical origins. The δ^15^N values of DIN from human and animal wastes have been found to be approximately 10–20‰, unlike the low values in atmospheric sources (2–8‰) and nitrogen-fixing cyanobacteria (0–2‰) [46]. Watanabe et al. [21] observed that the δ^15^N values in Manila clams were higher than those in POM, and they interpreted them as a reflection of DIN in coastal waters with anthropogenic nitrogen loading in tidal plates. Our results from the δ^15^N values of Manila clams in Korea expectedly demonstrated geographical differences (*p* < 0.05). The δ^15^N values of the four regions were 7.94 ± 1.19‰ (CC), 9.39 ± 0.59‰ (JB), 8.65 ± 0.82‰ (JN), and 10.02 ± 0.91‰ (GN). These nitrogen isotope ratios were in the range of those from previous studies conducted in the Jeonnam regions (Hwayang, Dolsan, and Jinahae) [19]. The differences observed the in Jeonbuk, Jeonnam, and Gyeongnam regions suggested that the river discharge causing the inflow of different DIN concentrations by season might have produced the ^15^N-enriched isotopic signature in clams [47].

Finally, the stable isotope ratios of these two elements were adopted for LDA (Table 2, Figure 3A) and a cross-validation procedure was used for evaluating the stability of the classification model. As a result, the predictability drawn from LDA was 84.0, 73.1, 60.0, and 53.8% for samples from CC, GN, JB, and JN, respectively. These results did not satisfactorily meet our discrimination rate goal. In particular, the clams from JN showed the lowest discrimination rate, where only 14 clams were appropriately identified and 12 clams were found to have other origins. Although previous studies have shown that the combination of δ^13^C and δ^15^N can be used for determining the geographical origins of agricultural products, plants, and meat [48,49], we have demonstrated the necessity of adding other factors that can increase the ability to identify habitats and other geographic information in food products, such as bivalves [19]. Given that the values of δD, δ^18^O, and δ^34^S measured in clams from the three regions of Korea did not demonstrate great discriminatory power against LDA [19], FAs were additionally selected in the next step of this study as a potential parameter for providing habitat information.

### 3.2. Second Step I: FA Composition for Geographical Authentication

FAs were added as another factor for better discrimination of samples that were not well identified by isotopic analysis, as their composition and contents vary depending on the habitat environment, and they are commonly analyzed in fisheries. This was done by selecting unauthenticated samples from the first LDA. The FA profile has been demonstrated to be a powerful indicator of geographic characteristics [2], and, specifically for our study, it is effective for tracing or understanding the habitat of shellfish [25,50]. As some FAs (e.g., C 20:5n-3, C18:1n-7) are mainly derived from diets, they have been considered as biomarkers for tracing the habitat or for habitat-based classification, considering shellfishes’ relationship to the food web [50].

Furthermore, the composition of FAs in plankton and its connected food web help elucidate the physiological conditions of the producer or their environment [51]. Gonçalves et al. [52] demonstrated that highly polyunsaturated FA (PUFA, e.g., C18:1n3) and unsaturated FA contents in phytoplankton can explain bloom episodes, as their highest contents can be measured under the condition of rapid cell growth, which requires high cell membrane fluidity. This indicates that the FA profile of organisms reflects their living environmental conditions, such as temperature, salinity, and food availability, as these factors can reflect the diet in the habitat [29].

In this study, five saturated FAs (SFAs), five monounsaturated FAs (MUFAs), and five PUFAs were analyzed (Appendix A). The FA concentrations showed similar patterns in each region, except for in some sampling sites (Appendix A). The relative composition of 15 FAs showed a high abundance of SFAs and specific patterns in several unsaturated FAs that possibly originated from primary producers. In our study, SFAs were the dominant FAs, representing 26.20–65.48% of all FAs. The major SFAs were palmitic (16:0) and stearic (18:0) acids. Palmitic acid is typically dominant in marine bivalves and can serve as a parameter of discrimination, as it varies with species, season, temperature, and diet [53]. PUFAs, which represented 10.23–32.33% of total FAs in our study, can also serve as markers for diet, as the most dominant PUFAs (eicosapentaenoic acid, EPA (20:5n-3) and docosahexaenoic acid, DHA (22:6n-3)) were not synthesized by clams but were mainly dependent on dietary input [36,53]. In a previous study, the ratio of PUFA to SFA (P/S) was used for identifying diet differences in clams collected from different regions in Korea [31], with the differences attributed to variations in the water temperature [29]. Similarly, FAs (such as 18:1n-9, 18:2n-6, 18:3n-3, 20:4n-6, 20:5n-3, and 22:6n-3) can act as biomarkers for dietary sources [28,54].

Our study showed differences in the composition of FAs that originated from diatoms (C20:5n3, C16:1n7), flagellates (C18:1n9, C22:6n3), and bacteria (C17:0, C18:1n7) (Figure 4, Appendix A). Among these FAs, the ratio of DHA to EPA (DHA/EPA) was >1 in samples from GN, indicating that clams from GN had more dinoflagellates in their diet. Parrish et al. [55] suggested that the ratio of DHA to EPA could be an indicator of diet, as this ratio was high in dinoflagellates and diatoms. This is consistent with the result that flagellates were dominant in GN-1 to GN-4, as per the annual report on plankton [56], and this phenomenon is believed to have influenced the grouping. Furthermore, GN uses a different aquaculture system (e.g., hanging method) for clams from other sites [57]; therefore, this culture condition could be the cause of the difference between clams from GN and those from other regions.

For the first additional step of authentication, three discriminant function axes were generated with LDA using FA profiles, and the sorted samples (*n* = 51) showed better discrimination results with respect to their geographical origins (Figure 3B, Table 3). The prediction results completely met our expectation (100%), indicating that data with similar values in dual-element stable isotopic ratios (C and N) can be used for classifying FA profiles and suggests improved authentication. We previously found that the multi-element isotopic analysis using C, N, and S to O, H, Sr, and Nd improved the reliability of the results, owing to the specific characteristics of each element [19]. Regarding FAs, the combination of SIA and FA analysis could be useful for the authentication of origin, for example, in distinguishing between farmed and wild Atlantic salmon (*Salmo salar*) [24]. Busetto et al. [58] reported the applicability of such a combination in determining the geographical origins of turbot (*Psetta maxima*) from different regions. The combined approach reported in previous studies, however, is based on all sample sets and therefore requires more time, samples, and expenditure for the results of each element and further analysis. This method differs from our approach, in that the stepwise approach employed in the present study focuses on less time- and labor-consuming methods for unclassified samples. Many studies have reported that FAs provide considerable information regarding diet, trophic transfer, and specific metabolic pathways [32] and show potential for authenticating such information in food, such as dairy products, shellfish, and livestock [59,60,61]. However, the analysis of FAs in a large number of samples is time-consuming [59], whereas this study demonstrated that supplementary experiments for determining FA profiles with small sample sizes are quite effective because they can be selectively applied to necessary samples. Furthermore, our data showed that FA profiles can be used for differentiating geographical origins after the first screening with basic stable isotopic analysis (C, and N).

### 3.3. Second Step II: Geographical Authentication Using CSIA of Carbon Isotopes in Fas

In this study, we evaluated geographical authentication using CSIA, a new promising technology that can provide habitat information based on producers, trophic relations, and metabolic pathways that might be affected by environmental conditions. The CSIA approach may overcome some of the limitations arising from bulk element isotopes that represent an integrated value analyzed in a sample [62]. This approach has been used in ecological and archaeological studies in recent years [63,64]. CSIA-FA can be utilized in various fields, such as food chain research, because it provides information on both FA composition and stable isotope ratios [32]. The δ^13^C values of FAs can provide a highly precise description of food sources and establish the traceability of carbon flow as consumer FA δ^13^C values varying based on the primary producer [32]. In food authentication, although the use of this method to certify aquaculture products (e.g., seafood) has been rarely explored, its potential has been anticipated, and it has enabled food certification for walnut oil or organic milk [65,66].

In this study, only two types of SFAs (16:0 and 18:0) and four MUFAs (18:1n-7, 18:1n-9, 20:1n-7, and 20:1n-9) were identified from the chromatogram (Table 4). The carbon isotopic values of Fas (δ^13^C-FA) in Manila clams from different regions showed significantly different patterns (*p* < 0.05), except for δ^13^C-FA 18:1n-7. Although no consistent or specific pattern was observed, the differences in values and profiles is believed to represent improved discrimination among the four regions. This result might be attributed to the different biological community structure of producers and the variation in metabolic states according to environmental conditions (e.g., temperature and salinity) in each region. In previous studies, for example, the environments of plankton communities that led to differences in FA profiles varied in latitude, geological characteristics, temperature, and type of land use [22,31].

Some FAs that cannot be synthesized by heterotrophic organisms and that originate from the diet can be reflected in clams as a diet source parameter. Therefore, predicting consumer diets on the basis of 16:0 and 18:0 can be problematic because most marine organisms contain these FAs. In this study, however, the discrepancy in isotopic values of C16:0 (palmitic acid)—which is ubiquitous as most organisms can synthesize it de novo—demonstrates the difference in the four regions’ environments. Furthermore, the difference in carbon isotope fractionation between stearic (18:0) and palmitic (16:0) acids in clams might be due to their habitat environment, as stearic acid is synthesized from palmitic acid through a series of elongation steps. The carbon isotope values of FAs serve a similar role to the bulk carbon stable isotope ratio in food sources and provide more specific information as a discriminator, as they reflect the dissolved inorganic carbon isotope values and fractionation through carbon assimilation [32]. As evidence for this assumption, Appendix A shows more than a 98% discrimination of geographical authentication after adding ^13^C-FA values to dual-element applications. However, the LDA results with δ^13^C-FA were not as well classified as the results obtained using FA profiles (Table 5; 58.3% in JB), possibly because several PUFAs that might have provided information on diet were excluded, because their isotope values could not be measured under GC-IRMS conditions due to their overlapping peaks. In other words, PUFAs that were not analyzed using the CSIA method could have played a significant role in determining habitat on the basis of diet, using the FA profile approach (Appendix A).

### 3.4. Applications for Imported Clams: Geographical Authentication Using Carbon Isotopes of FAs from Small Amounts of Samples

Manila clams from China and DPR Korea were additionally included for evaluating their geographical origins using the proposed stepwise method (Figure 1 and Figure 2). The stable isotope ratios of the carbon and nitrogen contents and carbon stable isotopes of the FAs are listed in Table 6, Figure 5 and Appendix A.

The average δ^13^C values of *R. philippinarum* sampled in China, DPR Korea, and Korea were −17.58 ± 0.3, −17.69 ± 0.38, and −16.94 ± 0.79‰, respectively, which were statistically different (*p* < 0.05) from each other. The δ^13^C values of the clams from China are similar to the those for the Japanese scallop *Patinopecten yessoensis* (−17.65 ± 0.18‰) collected from Penglai province (Shandong) near the sampling site [67] and to the range of δ^13^C values of POM reported by Wu et al. [68]. The δ^13^C values of the clam samples from DPR Korea were slightly lower than those of the samples from Korea. In this study, the δ^15^N values of Manila clams from the three countries showed similar ranges (China: 9.67 ± 0.72‰; DPR Korea: 8.95 ± 0.24‰; Korea: 9.01 ± 1.18‰, *p* > 0.05). LDA performed on all samples (*n* = 133) showed well-grouped results (Figure 6A). However, the results from the cross-validation procedure showed that clams from the three countries could not be clearly distinguished (average: 77.44%), with low predictability found in the clams from China and DPR Korea (CHN: 26.7%, KOR: 96.1%, NK: 6.3%). However, the classification rate of samples from Korea was correctly discriminated with respect to geographical origin.

This means that, although it is necessary to evaluate the reliability of the applied method, a simple method can be sufficiently useful depending on which characteristic needs to be determined, such as distinguishing only imported or domestic products. However, this result also indicates that the stable isotope ratios of two elements, C and N, were inadequate for determining the geographical origin among local regions of Korea but sufficient for discriminating between clams of Korea, China, and DPR Korea.

For better authentication, the FA profiles and their carbon isotopic ratios were separately applied as an additional step. This is because certain compounds, such as C18:1n-9, C20:1n-7, and C20:1n-9, can be used as specific biomarkers for dinoflagellates, microalgae, and copepods, respectively [28]. They can also be used in food research for determining environmental conditions, although it is worth noting that planktonic composition (community) was observed to vary seasonally and regionally [31,53]. The regional studies described in Section 3.2 on the Korean clams indicated the possibility that such parameters can be used in this study for tracing geographical origins via the FA profiles of the intertidal zone plankton communities adjacent to sites where the clams were collected [56].

Seasonal changes in dinoflagellate abundance were observed in the southern sea (Korea), and the difference in dominance of diatoms and dinoflagellates in the western and southern parts of Korea supports the differences in FA profiles in our samples from local habitats [56,69]. In addition, the FA profile of clams from China is similar to that reported in previous studies, consistent with the predominance of diatoms over flagellates [70]. The FA profiles of clams from China, DPR Korea, and Korea were significantly different in several compounds, including C15:0, C18:0, C16:1n-7, C18:1n-7, C18:2n-6, and C20:5n-3 (Appendix A, *p* < 0.05). The observed differences could be due to the characteristics of their environmental conditions and seasonal changes in the food sources in the surrounding water. However, the cross-validation model with FA profiles demonstrated a relatively low (98.24%) discrimination ratio (Table 7). Unlike in the study of regional differences in Korea (Section 3.2), the FA profiles were not sufficient for discriminating the country of origin based on the cross-validated data. However, simple LDA (not cross-validated) showed good separation among the groups of clams from the three countries (Figure 6B).

Finally, the stable isotope ratios of biomarker FAs were considered. The δ^13^C values of C18:1n-7 were significantly different between the clams from the three countries (Appendix A, *p* < 0.05, China: −16.52 ± 0.44‰, DPR Korea: −21.18 ± 0.45‰, Korea: −22.69 ± 2.15‰), and considering the FA profiles, C18:1n-7 was an important factor for differentiating the clams as its content was significantly different between each country. This indicated the ability of δ^13^C values of C18:1n-7 to determine origins. The highest MUFA content was detected in samples from DPR Korea. As predicted, statistical analyses showed perfect separation of the samples, even based on their cross-validated data (Figure 6C and Table 7).

## 4. Conclusions

Considering the ambiguity of standards of the origins of seafood, information on the habitats of aquatic products is important. Many studies have reported methods for geographical tracing, but most of these studies have focused only on identifying parameters that can be used effectively. Stable isotope analysis is a possible approach for determining geographical differences. In many studies, for example, the mapping of isotope values is considered a good method for understanding the environment or origins of elements. However, it is difficult to distinguish the geographical origins of certain samples, such as those in our recent study, where multi-element isotopes have been applied for food authentication. However, even when using seven elements (C, N, S, O, D, Sr, and Nd), the bulk isotope did not show a discrimination rate of >90%, and one element (Sr) even lowered the discrimination power [19].

In this study, the FA profiles and their stable carbon isotope ratios were used as additional parameters for overcoming this limitation of bulk isotope analysis of enlarged element numbers for the analytes. In particular, FA profiles and compound-specific carbon isotopic values can provide useful information for determining the geographic origin based on diet even in samples with similar isotope values. The results of our study additionally indicate that the FA profiles and δ^13^C-FAs can serve as a potential tool for tracing shellfish, even among small sample sizes. This stepwise approach can be used appropriately for processing enormous samples or for samples that require different degrees of discrimination.

In addition, the combination of each step can be considered for further applications (Appendix A). With this experiment surpassing our objectives, its results have significant relevance for the advancement of food authentication using stable isotope analysis and compound-specific isotopic analysis. The results showed that all classification models combined with δ^13^C-FAs obtained an overall accurate classification rate of 100% (Appendix A). This means that the six different specimens were not only well separated from the domestic samples (Korea) but were also properly classified into separate groups based on country (China and DPR Korea). In our recent work with stable isotope ratios, a combination of multi-isotopes from elements showed an increased classification rate in identifying the geographical origin of Manila clams compared to when adopting single or less than three elements [19]. Similarly, this study demonstrated the advantage of using several parameters simultaneously to identify geographic origins over protocols that use individual analytical methods. These satisfactory results show the potential of δ^13^C, δ^15^N, and δ^13^C-FA for building a Manila clam identification database and indicate that the stepwise application of δ^13^C, δ^15^N, and δ^13^C-FAs by the way of additional analysis or sequential approaches may enhance the prediction capabilities of tracing the origin of Manila clams in Korea.

## Figures and Tables

**Figure 1 foods-11-01965-f001:**
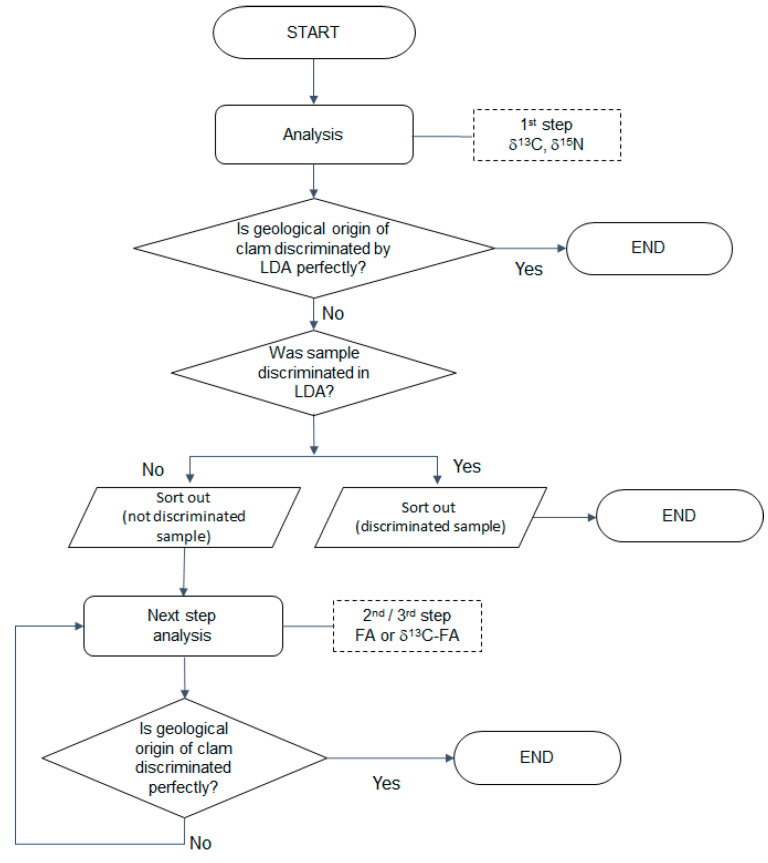
A schematic flow diagram of stepwise application of dual-element stable isotope analysis, fatty acid composition (FA), and compound-specific isotope analysis (CSIA) of FAs (δ^13^C-FA). This process enables the collection of geographic information included in each parameter and application of this information for determining geographical origin.

**Figure 2 foods-11-01965-f002:**
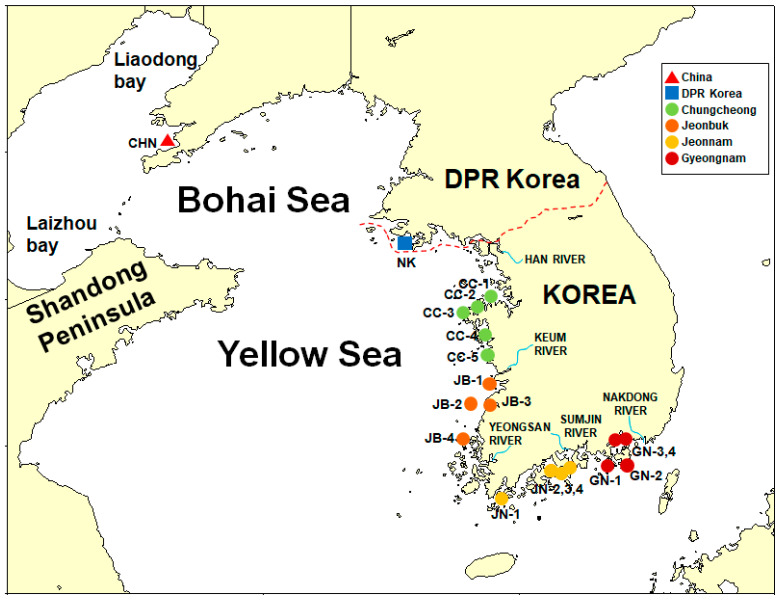
Regions of three countries from where samples of the clam *Ruditapes philippinarum* were collected. The triangle, square, and circles indicate the collection sites, that is, the sites of origin from China (*n* = 15), the Democratic People’s Republic of Korea (*n* = 16), and Korea (*n* = 102), respectively. The collection sites in Korea are differentiated according to the provinces: Chungcheong (CC; *n* = 25), Jeonbuk (JB; *n* = 25), Jeonnam (JN; *n* = 26), and Gyeongnam (GN; *n* = 26).

**Figure 3 foods-11-01965-f003:**
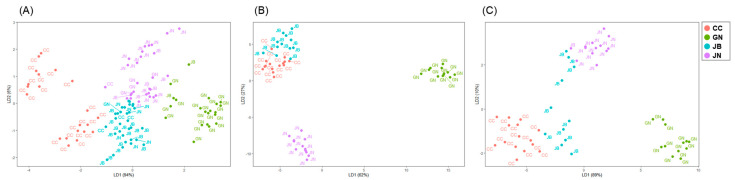
Linear discriminant analysis of Manila clams. (**A**–**C**) shows the LDA score plot of *Ruditapes philippinarum* collected from four different regions of Korea. The four colors represent samples from different regions, and the figure shows samples that were not well identified by the different colors at each site. In each case, stepwise application of LDA was performed for (**A**) dual-element isotopes, (**B**) fatty acid (FA) profiles, and (**C**) δ^13^C−FAs. Each color indicates Chungcheong (CC), Gyeongnam (GN), Jeonbuk (JB), and Jeonnam (JN), respectively.

**Figure 4 foods-11-01965-f004:**
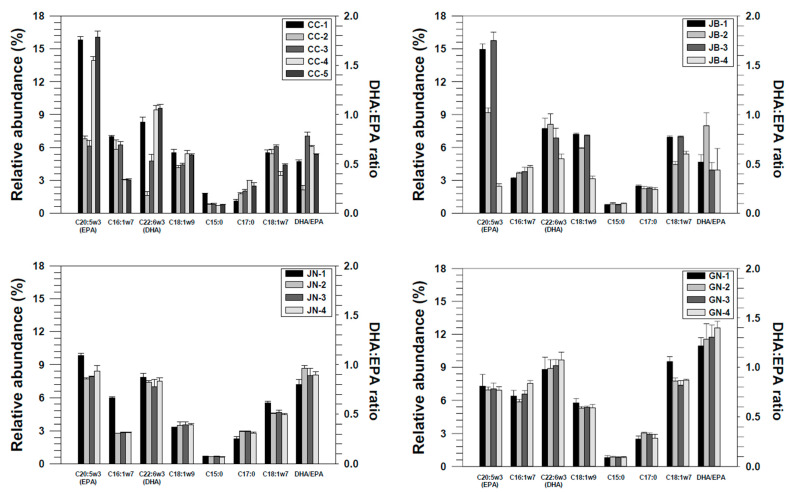
Relative abundance of biomarker fatty acids (diatom, flagellate, and bacteria) and the ratio of DHA to EPA in Manila clams from four different regions (Chungcheong (CC), Jeonbuk (JB), Jeonnam (JN), and Gyeongnam (GN)) of Korea for evaluating the source differences.

**Figure 5 foods-11-01965-f005:**
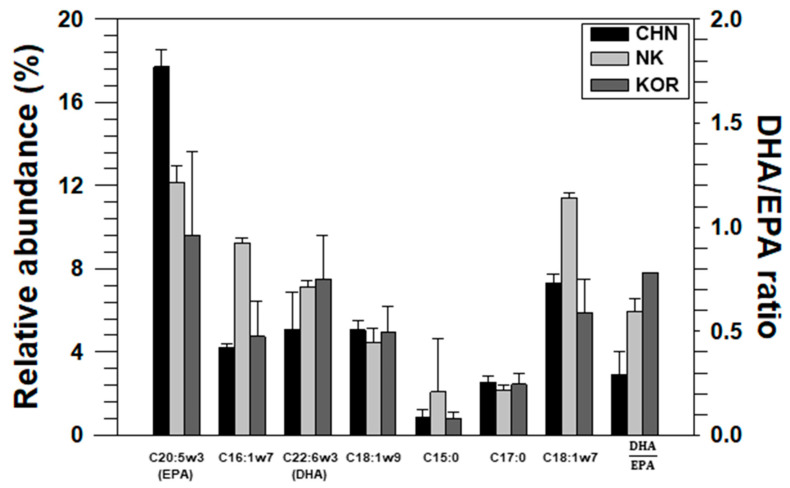
Relative abundance of biomarker fatty acids (diatom, flagellate, and bacteria) and the ratio of DHA to EPA in Manila clams from Korea (*n* = 51), China (*n* = 3), and DPR Korea (*n* = 3).

**Figure 6 foods-11-01965-f006:**
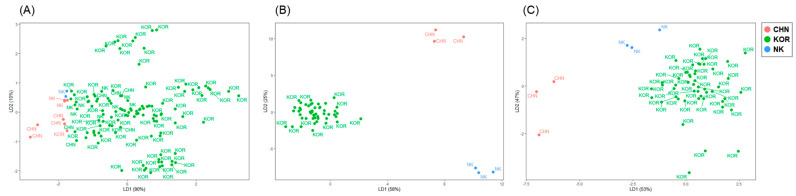
Linear discriminant analysis of Manila clams. (**A−C**) shows the LDA score plot of Manila clams collected from Korea, DPR Korea, and China. Each case shows the stepwise application of LDA for (**A**) dual−element isotopes, (**B**) fatty acid profiles, and (**C**) δ^13^C−FAs. Each color indicates clams from China (CHN), Korea (KOR), and DPR Korea (NK).

**Table 1 foods-11-01965-t001:** Isotopic values (δ13C, and δ15N) of the Manila clam *Ruditapes philippinarum* from various provinces of Korea. Different letters indicate significant differences with respect to regions (ANOVA, Tukey’s, *p* < 0.05).

Region	ID	δ^13^C (‰)	δ^15^N (‰)	Remarks (Sites, Sample N)	Sampling Date
Chungcheong	CC-1	−18.0 ± 0.2	8.9 ± 0.2	Dangjin (37.035906° N, 126.573153° E, *n* = 5)	21 July 2016
(*n* = 25)	CC-2	−18.3 ± 0.3	6.8 ± 0.3	Sogeunri (36.816236° N, 126.144142° E, *n* = 5)	21 June 2016
	CC-3	−18.4 ± 0.1	6.4 ± 0.4	Uihang (36.887164° N, 126.372181° E, *n* = 5)	21 June 2016
	CC-4	−17.6 ± 0.4	9.1 ± 0.2	Hongseong (36.519644° N, 126.484239° E, *n* = 5)	22 June 2016
	CC-5	−17.5 ± 0.3	8.6 ± 0.4	Boryeong (36.232231° N, 126,509742° E, *n* = 5)	23 June 2016
	Average	−18.0 ± 0.4 ^a^	7.9 ± 1.2 ^a^		
Jeonbuk	JB-1	−17.1 ± 0.1	9.6 ± 0.4	Saemangeum (35.836264° N, 126.544094° E, *n* = 10)	24 June 2016
(*n* = 25)	JB-2	−16.6 ± 0.3	9.1 ± 0.3	Wedo (35.576103° N, 126.254097° E, *n* = 5)	21 June 2016
	JB-3	−17.4 ± 0.1	10.0 ± 0.3	Gochang (35.549975° N, 126.559472° E, *n* = 5)	24 June 2016
	JB-4	−16.4 ± 0.4	8.7 ± 0.4	Shinan (35.103347° N, 126.1363342° E, *n* = 5)	20 July 2016
	Average	−16.9 ± 0.4 ^b^	9.4 ± 0.6 ^c^		
Jeonnam	JN-1	−16.5 ± 0.3	7.7 ± 0.1	Wando (34.278747° N, 126.729958° E, *n* = 10)	14 June 2016
(*n* = 26)	JN-2	−17.1 ± 0.1	10.0 ± 0.2	Hwayang (34.688503° N, 127.600969° E, *n* = 3)	24 July 2015
	JN-3	−16.9 ± 0.2	9.0 ± 0.1	Dolsan (34.669314° N, 127.766706° E, *n* = 3)	29 July 2015
	JN-4	−16.9 ± 0.2	9.1 ± 0.1	Yeosu (34.614800° N, 237.676294° E, *n* = 10)	26 May 2016
	Average	−16.8 ± 0.3 ^b^	8.7 ± 0.8 ^b^		
Gyeongnam	GN-1	−16.8 ± 0.3	8.8 ± 0.2	Tongyeong (34.733336° N, 128.386897° E, *n* = 8)	14 July 2016
(*n* = 26)	GN-2	−15.7 ± 0.1	10.6 ± 0.2	Geoje (34.732281° N, 128.691311° E, *n* = 10)	20 May 2016
	GN-3	−16.3 ± 0.1	9.8 ± 0.2	Masan (35.096711° N, 128.672411° E, *n* = 3)	24 July 2015
	GN-4	−15.8 ± 0.2	11.0 ± 0.1	Udo (35.087564° N, 128.720750° E, *n* = 5)	15 July 2016
	Average	−16.1 ± 0.6 ^c^	10.0 ± 0.9 ^c^		

**Table 2 foods-11-01965-t002:** Predicted geographical origins of each clam from different provinces of Korea. The number of individual clams with geographical origins defined by linear discriminant analysis (LDA) of C–N stable isotope combinations. Each number represents a sample classified by that predicted region, and the prediction was calculated using the number of samples (bolds) whose origin was well identified out of the total samples. The results were evaluated using cross-validation.

Region	Number of Samples Classified by LDA				Prediction
	CC	GN	JB	JN	(%)
Chungcheong (CC)	**21**	0	3	1	84.0
Gyeongnam (GN)	0	**19**	2	5	73.1
Jeonbuk (JB)	0	2	**15**	8	60.0
Jeonnam (JN)	0	0	12	**14**	53.8

**Table 3 foods-11-01965-t003:** Predicted geographical origins of clams from different provinces of Korea. Number of individual clams with geographical origins determined by LDA of fatty acid profiles. Each number represents a sample classified according to the predicted region, and prediction was performed using the number of samples (bolds) whose origin was well identified out of the total number of samples. The results are evaluated using cross-validation.

Region	Number of Samples Classified by LDA				Prediction
	CC	GN	JB	JN	(%)
Chungcheong (CC)	**15**	0	0	0	100
Gyeongnam (GN)	0	**12**	0	0	100
Jeonbuk (JB)	0	0	**12**	0	100
Jeonnam (JN)	0	0	0	**12**	100

**Table 4 foods-11-01965-t004:** Carbon isotopic values (‰) of FAs of the Manila clam *Ruditapes philippinarum* with different geographical origins. Each value was represented as the average isotope ratios (±SD) of three samples. Different letters indicate significant differences in δ^13^C-FAs with respect to regions (ANOVA, Tukey’s, *p* < 0.05).

Regions	ID	δ^13^C16:0	δ^13^C18:0	δ^13^C18:1n-7	δ^13^C18:1n-9	δ^13^C20:1n-7	δ^13^C20:1n-9
Chungcheong	CC-1	−23.1 ± 0.4	−24.6 ± 0.4	−20.9 ± 0.8	−15.2 ± 0.7	−22.9 ± 0.5	−21.0 ± 0.7
(*n* = 15)	CC-2	−24.4 ± 0.4	−25.3 ± 0.4	−22.5 ± 0.4	−15.6 ± 0.5	−25.0 ± 0.3	−22.4 ± 0.3
	CC-3	−24.4 ± 0.4	−25.3 ± 0.4	−22.2 ± 0.3	−15.5 ± 0.4	−24.6 ± 0.3	−21.3 ± 0.5
	CC-4	−25.0 ± 0.5	−25.3 ± 0.3	−21.1 ± 0.7	−18.5 ± 0.5	−20.7 ± 0.5	−19.9 ± 0.3
	CC-5	−24.9 ± 0.4	−25.0 ± 0.3	−20.2 ± 0.4	−17.9 ± 0.4	−20.6 ± 0.6	−20.4 ± 0.5
	Avg.	−24.3 ± 0.8 ^a^	−25.1 ± 0.4 ^a^	−21.4 ± 1.0 ^b^	−16.5 ± 1.5 ^b^	−22.8 ± 2.0 ^b^	−21.0 ± 1.0 ^bc^
Jeonbuk	JB-1	−20.4 ± 0.5	−24.5 ± 0.4	−20.6 ± 0.5	−15.6 ± 0.5	−21.0 ± 0.4	−20.9 ± 0.4
(*n* = 12)	JB-2	−21.9 ± 0.4	−21.4 ± 0.8	−21.4 ± 0.5	−15.0 ± 0.7	−21.1 ± 0.9	−21.6 ± 0.5
	JB-3	−26.8 ± 0.4	−26.1 ± 0.8	−24.5 ± 0.7	−19.6 ± 0.5	−24.8 ± 0.6	−21.7 ± 0.6
	JB-4	−23.5 ± 0.4	−21.2 ± 0.3	−21.2 ± 0.4	−17.5 ± 0.5	−21.5 ± 0.4	−23.6 ± 0.5
	Avg.	−23.1 ± 2.5 ^ab^	−23.3 ± 2.2 ^b^	−21.9 ± 1.7 ^b^	−16.9 ± 1.9 ^ab^	−22.1 ± 1.7 ^bc^	−21.9 ± 1.1 ^ab^
Jeonnam	JN-1	−22.2 ± 0.4	−19.8 ± 0.3	−20.7 ± 0.5	−16.8 ± 0.5	−20.6 ± 0.5	−21.3 ± 0.5
(*n* = 12)	JN-2	−23.2 ± 0.3	−19.8 ± 0.4	−22.0 ± 0.5	−18.0 ± 0.4	−21.5 ± 0.5	−23.1 ± 0.5
	JN-3	−23.4 ± 0.4	−21.2 ± 0.6	−22.6 ± 0.5	−18.0 ± 0.4	−20.9 ± 0.6	−23.3 ± 0.4
	JN-4	−23.5 ± 0.3	−20.6 ± 0.5	−22.1 ± 0.6	−18.0 ± 0.4	−21.8 ± 0.4	−23.4 ± 0.5
	Avg.	−23.1 ± 0.6 ^ab^	−20.3 ± 0.7 ^c^	−21.9 ± 0.9 ^b^	−17.7 ± 0.7 ^ab^	−21.2 ± 0.7 ^c^	−22.8 ± 1.0 ^a^
Gyeongnam	GN-1	−24.5 ± 0.6	−24.2 ± 0.6	−27.1 ± 0.7	−19.5 ± 0.6	−26.1 ± 0.5	−23.6 ± 0.6
(*n* = 12)	GN-2	−21.6 ± 0.5	−22.1 ± 0.3	−25.6 ± 0.6	−17.8 ± 0.6	−24.6 ± 0.5	−18.0 ± 0.6
	GN-3	−21.9 ± 0.3	−22.2 ± 0.4	−25.5 ± 0.6	−17.8 ± 0.3	−24.8 ± 0.3	−18.0 ± 0.5
	GN-4	−22.0 ± 0.5	−22.2 ± 0.7	−25.7 ± 0.5	−17.8 ± 0.4	−24.8 ± 0.5	−18.1 ± 0.3
	Avg.	−22.5 ± 1.3 ^b^	−22.7 ± 1.0 ^b^	−26.0 ± 0.9 ^a^	−18.2 ± 0.9 ^a^	−25.1 ± 0.7 ^a^	−19.4 ± 2.6 ^c^

**Table 5 foods-11-01965-t005:** Predicted geographical origins of clams from different provinces of Korea. Numbers of individual clams with geographical origins determined by LDA of carbon isotope ratios of fatty acids (δ^13^C-FA). Each number represents a sample classified according to the predicted region; prediction was performed using the number of samples (bolds) whose origin was well-identified out of the total number of samples. The results are evaluated using cross-validation.

Region	Number of Samples Classified by LDA (Cross-Validation)				Prediction
	CC	GN	JB	JN	(%)
Chungcheong (CC)	**14**	0	1	0	93.3
Gyeongnam (GN)	0	**12**	0	0	100
Jeonbuk (JB)	1	0	**7**	4	58.3
Jeonnam (JN)	0	0	0	**12**	100

**Table 6 foods-11-01965-t006:** Isotopic values (δ^13^C and δ^15^N) of the Manila clam *Ruditapes philippinarum* from Korea, China, and DPR Korea. Different letters indicate significant differences of isotope values according to countries (ANOVA, Tukey’s, *p* < 0.05). * The sampling dates from China and DPR Korea indicate the dates they were acquired in the process of importation into Korea.

Country	ID	δ^13^C (‰)	δ^15^N (‰)	Sampling Date *
China (Dalian)	CHN	−17.58 ± 0.39 ^b^	9.67 ± 0.72 ^a^	1 September 2015
DPR Korea	NK	−17.69 ± 0.38 ^b^	8.95 ± 0.24 ^a^	1 September 2015
Korea	KOR	−16.94 ± 0.79 ^a^	9.01 ± 1.18 ^a^	-

**Table 7 foods-11-01965-t007:** Number of individual clams with geographical origins determined using LDA with a stepwise approach. The numbers indicate the samples whose origin was well identified among the samples from each country (China, *n* = 3; DPR Korea, *n* = 3; Korea, *n* = 51). All data are cross-validation results for each geographical origin using the leave-one-out cross-validation (LOOCV) method.

Country	Number of Samples Correctly Classified		
	Step 1. δ^13^C, δ^15^N	Step 2-I. FA Profiles	Step 2-II. δ^13^C-FA
China (CHN)	15 (4)	3 (3)	3 (3)
Korea (KOR)	102 (98)	51 (51)	51 (51)
DPR Korea (NK)	16 (1)	3 (2)	3 (3)
Total (Cross-Val)%	77.44%	98.24%	100%

## Data Availability

The data presented in this study are available on request from the corresponding author.

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
