# Peer review of "Stepwise Approach for Tracing the Geographical Origins of the Manila Clam Ruditapes philippinarum Using Dual-Element Isotopes and Carbon Isotopes of Fatty Acids"

_foods, 2022, doi:10.3390/foods11131965_

Round 1
Reviewer 1 Report
Line 129: Please correct the cited reference no 17 ! In text is Magdas et al., 2021, and in Reference section is Won et al., 2021.
Through the manuscript, please modified all isotopic values to one decimal (you have to round all data to one decimal). Usually, in published articles, the isotopic values are presented having only a decimal.
Line 529: You mentioned six elements {even when six elements (C, N, S, O, D, Sr, and Nd)}, but there are seven, please verify and make the correction.
Author Response
Line 129: Please correct the cited reference no 17 ! In text is Magdas et al., 2021, and in Reference section is Won et al., 2021.
- Two studies were used as references in this sentence, but one was missing in the process of revising the format. We added “Magdas et al., 2021” in the body and the list of reference as [33]. Thank you for your careful review.
Through the manuscript, please modified all isotopic values to one decimal (you have to round all data to one decimal). Usually, in published articles, the isotopic values are presented having only a decimal.
- We corrected it. Thank you.
Line 529: You mentioned six elements even when six elements (C, N, S, O, D, Sr, and Nd), but there are seven, please verify and make the correction.
- In fact, it was discussed that the combination of 6 elements showed greater differentiation than the combination using 7 elements in the previous study (Won et al., 2021). So, the sentence has been corrected to reduce misunderstandings as follow.
Revision)
In addition, even when using seven elements (C, N, S, O, D, Sr, and Nd), the bulk isotope did not show more than 90% discrimination rate, and one element (Sr) even lowered the discrimination power

Reviewer 2 Report
Before an in-depth review of this manuscript is possible, some relevant questions need to be answered:
Row 121: Which "hundreds of samples" are you talking about here?
Row 133: Give exact sampling dates.
Row 137: Which interval describes "long-term dietary information"? Explain in detail. Is this interval identical for SIA, FA and CSIA?
Compare your present results with your earlier (2015) data (your reference 17). How do they fit? How do you explain the differing d34S-results?
Fig. 2: Give detail maps for the 4 regions in Korea where the samples have been collected. Add the sample points of the samples collected 2015, if different, or explain which present sample points they represent. Also add detail maps for NK and China, if earlier sample points differ from present ones.
Rows 150-151: Explain defatting in more detail.
Table 1: Give also minimum and maximum values. Integrate Table S3 into Table 1. Add a table (or several tables) in the supplements containing all results for every individual sample.
Rows 279-280: This explanation is incorrect.
Table 2: add NK and China data, or produce additional table also containing these data.
Fig. 4: Add NK and China data, or produce additional figures also containing these data.
Table 3, 4 and 5: Add NK and China data (Table S5 in Table 4, etc..).
Row 510: Figure numbering incorrect - Figure 3 is around Row 305. Still, I think it will be good to combine these figures (to show them at the same place) that comparison of figures is easier. Also, show the regions abbreviations instead of KOR (in comparison of countries).
Conclusions: It is good that, at last, you do mention your previous results, however, as already written above, I think it is necessary that the previous results are in detail discussed before.
Table S1: Add NK and China data. Add statistical evaluation of significant difference. Add unit(s).
Where are the CSIA-data (NK, China)?
Minor points:
Row 71: daily?
Row 74: Is SIA really a novel method? our reference 14 dates back more than 30 years.
Row 129: reference mentioned misses in reference list, number refers to other reference. Check references!
Table 6: Step 3
Row 530: Rewrite, I think you want to have another meaning.
Author Response
Row 121: Which "hundreds of samples" are you talking about here?
-> We revised the sentence more clearly.
Revision)
For this, samples from different sites in Korea (section 2.1) were first applied, and we adopted this stepwise approach for selected samples from two adjacent countries (China and Democratic People’s Republic Korea (DPR Korea)).
Row 133: Give exact sampling dates.
-> We added exact sampling information (sampling date and site (latitude/longitude)) in the result table (Table 1).
Row 137: Which interval describes "long-term dietary information"? Explain in detail. Is this interval identical for SIA, FA and CSIA?
-> Long-term information refers to the turnover rate of the compounds (carbon, nitrogen, fatty acids) of interest in this study, which are intended to be used to evaluate their diet or habitat information. Adductor muscle have slow turnover rate compared to another organ. We revised the sentence in the manuscript add more references.
Revision)
After collecting samples from the sites, the samples were brought to the laboratory, and the adductor muscle was dissected and used as an appropriate target organ for isotope analysis for obtaining long-term dietary information as adductor muscle have slower turnover and much lower lipid contents than other tissues [22].
Compare your present results with your earlier (2015) data (your reference 17). How do they fit? How do you explain the differing d34S-results?
-> Thank you very much for your valuable comments, which are important to help us improve the manuscript quality. First of all, the results of carbon and nitrogen stable isotopes were a range that included values from the previous study as well as values from adjacent regions. We discuss this results in the revised manuscript. However, regarding d34S, we found a difference in isotope values ​​within 2‰ between current study and the previous one. In fact, in this study, the stable isotope composition of d34S was measured by an analytical service company in UK. Although this study used a different set of samples with the previous one, we decided that additional checks should be needed to be better confirmation because the current samples were analyzed by a different institution. In particular, the results of the previous study (Won et al., 2021) had already confirmed the reliability of chromatography during this revision process. So, cross-validation will be necessary for d34S. However, in this study, the discriminant power of d34S is little significant. So, the sulfur data was excluded in the revised manuscript because we focused on CSIA as an additional step for food origin certification. All results have been recalculated (there was no significant changes) and thoroughly corrected. Thanks again for your helpful comments.
Fig. 2: Give detail maps for the 4 regions in Korea where the samples have been collected. Add the sample points of the samples collected 2015, if different, or explain which present sample points they represent. Also add detail maps for NK and China, if earlier sample points differ from present ones.
-> Both the sites (latitude and longitude) and their IDs are annotated in Figure 2 and revised Table 1. Additionally, except for two sites (Hwayang (JN-2 for this study and YH for previous study) and Dolsan (JN-3, YD)), the sampling points are different from previous study, and even different set of samples was used for all sites (different sampling date). In this study, we did not consider adding sites of previous study to the map. However, we agree that comparative discussion with previous studies is meaningful, as suggested by the reviewer. Therefore, the sampling points are mentioned by adding a paragraph that compares and discusses the overlapping part with the previous paper.
Rows 150-151: Explain defatting in more detail.
-> We added detailed information about removing fat for carbon isotope analysis with the reference.
Revision)
Additionally, lipids, which exhibit depleted isotopic values, were removed using a mixture of chloroform and methanol (2:1, v/v ratio) as described in Kim et al. [20]. In brief, a mixed solvent was added to the sample and shaken to remove the lipids layer separated on the supernatant and this step was repeated three times.
Table 1: Give also minimum and maximum values. Integrate Table S3 into Table 1. Add a table (or several tables) in the supplements containing all results for every individual samples.
-> The minimum and maximum values of the isotopic composition have been incorporated into the Table S3 but two tables are not integrated in the main text as suggested by the reviewer because the raw data set was more than one hundred. Please found Table S3 for their raw data with their minimum and maximum data.
Rows 279-280: This explanation is incorrect.
-> The sentence was deleted, thank you.
Table 2: add NK and China data, or produce additional table also containing these data.
Fig. 4: Add NK and China data, or produce additional figures also containing these data.
Table 3, 4 and 5: Add NK and China data (Table S5 in Table 4, etc..).
-> We added new table and figure (Table 6, Fig.6) for the NK and China data of isotopic ratios and fatty acid composition, respectively. We try to reduce the repetition in the listing results. We believe that newly added table and figure with the last supplementary materials can cover the all the comments review mentioned. Thank you.
Row 510: Figure numbering incorrect - Figure 3 is around Row 305. Still, I think it will be good to combine these figures (to show them at the same place) that comparison of figures is easier. Also, show the regions abbreviations instead of KOR (in comparison of countries).
-> We revised the figure number in the manuscript. However, since the two figures (Figure 3 and Figure 5) are based on different samples (Figure 3 is Korean sample and Figure 5 is for three countries), we have separated them to place in different paragraphs.
Conclusions: It is good that, at last, you do mention your previous results, however, as already written above, I think it is necessary that the previous results are in detail discussed before.
-> Thank you, we agree to the reviewer opinion. We considered discussing the previous study and also revised the manuscript for emphasizing new approach in this study.
Revision)
However, it is difficult to distinguish the geographical origins of some samples. In our recent study, for example, multi-element isotopes have been applied for food authentication. However, In this study, the FA profiles and their stable carbon isotope ratios were used as additional parameters for overcoming the limitation of bulk isotope analysis of enlarged element numbers. In particular, FA profiles and compound-specific carbon isotopic values can provide useful information for determining the geographic origin, even in samples with similar isotope values, on the basis of diet.
Table S1: Add NK and China data. Add statistical evaluation of significant difference. Add unit(s).
Where are the CSIA-data (NK, China)?
-> First, the data for NK and China were listed in Table S3, Table S4, and newly added table and figure (Table 6, Fig. 6). The unit and the results of statistical significances (ANOVA, P<0.05) also marked with letters in table (Table S3) with raw data. As we divided the main text into two paragraphs, one to show the stepwise approach and the other for application, the results on China and NK are presented as second paragraphs of main text.
Minor points:
Row 71: daily?
-> Thank you, we corrected it.
Row 74: Is SIA really a novel method? our reference 14 dates back more than 30 years.
-> We change the word and also try to emphasize the CSIA rather than SIA.
Revision)
- Stable isotope analysis (SIA) is a good method for determining environmental information, habitat, and ecosystem-based diet [16,17].
- In addition, compound-specific isotope analysis (CSIA), which has been recently gaining attention, can provide additional specific information on the habitat-and diet-derived FA patterns and carbon sources of an organism (e.g., d13C-FA) [32].
Row 129: reference mentioned misses in reference list, number refers to other reference. Check references!
-> We accidentally missed it when formatting but revised. Thank you.
Table 6: Step 3
-> In this study, the fatty acid composition and isotopes of fatty acid were set as factor can be applied separately. So, each step marked as step2-I and step2-II in the revised manuscript to reduce misunderstanding.
Row 530: Rewrite, I think you want to have another meaning
-> We revised the sentences. Thank you.
Revision)
In addition, even when using seven elements (C, N, S, O, D, Sr, and Nd), the bulk isotope did not show more than 90% discrimination rate, and one element (Sr) even lowered the discrimination power
